# Long-Term Shaping of Corticostriatal Synaptic Activity by Acute Fasting

**DOI:** 10.3390/ijms22041916

**Published:** 2021-02-15

**Authors:** Federica Campanelli, Daniela Laricchiuta, Giuseppina Natale, Gioia Marino, Valeria Calabrese, Barbara Picconi, Laura Petrosini, Paolo Calabresi, Veronica Ghiglieri

**Affiliations:** 1Dipartmento di Medicina, Università di Perugia, 06129 Perugia, Italy; federica.campanelli@unicatt.it (F.C.); giuseppina.natale@studenti.unipg.it (G.N.); gioia.marino@gmail.com (G.M.); valeria.calabrese@sanraffaele.it (V.C.); 2Dipartimento di Neuroscienze, Facoltà di Medicina e Chirurgia, Università Cattolica del Sacro Cuore, 00168 Rome, Italy; paolo.calabresi@policlinicogemelli.it; 3Laboratorio di Neurofisiologia Sperimentale e del Comportamento, IRCCS Fondazione Santa Lucia c/o CERC, 00143 Rome, Italy; daniela.laricchiuta@gmail.com (D.L.); laura.petrosini@uniroma1.it (L.P.); 4IRCCS San Raffaele Pisana, Rome 00176, Italy; barbara.picconi@uniroma5.it; 5Università Telematica San Raffaele, 00166 Rome, Italy; 6Clinica Neurologica, Fondazione Policlinico Universitario Agostino Gemelli IRCCS, 00168 Rome, Italy

**Keywords:** food deprivation, dietary restriction, dorsolateral striatum, GluA1, calcium-permeable AMPA, naphthyl-acetyl spermine

## Abstract

Food restriction is a robust nongenic, nonsurgical and nonpharmacologic intervention known to improve health and extend lifespan in various species. Food is considered the most essential and frequently consumed natural reward, and current observations have demonstrated homeostatic responses and neuroadaptations to sustained intermittent or chronic deprivation. Results obtained to date indicate that food deprivation affects glutamatergic synapses, favoring the insertion of GluA2-lacking α-Ammino-3-idrossi-5-Metil-4-idrossazol-Propionic Acid receptors (AMPARs) in postsynaptic membranes. Despite an increasing number of studies pointing towards specific changes in response to dietary restrictions in brain regions, such as the nucleus accumbens and hippocampus, none have investigated the long-term effects of such practice in the dorsal striatum. This basal ganglia nucleus is involved in habit formation and in eating behavior, especially that based on dopaminergic control of motivation for food in both humans and animals. Here, we explored whether we could retrieve long-term signs of changes in AMPARs subunit composition in dorsal striatal neurons of mice acutely deprived for 12 hours/day for two consecutive days by analyzing glutamatergic neurotransmission and the principal forms of dopamine and glutamate-dependent synaptic plasticity. Overall, our data show that a moderate food deprivation in experimental animals is a salient event mirrored by a series of neuroadaptations and suggest that dietary restriction may be determinant in shaping striatal synaptic plasticity in the physiological state.

## 1. Introduction

Dietary restriction and acute food deprivation (fasting) are voluntary practices currently used in many cultures for religious and health reasons [1,2,3]. The popularity of abstaining habits relies on their durable beneficial effects at a systemic level. In humans, fasting has effects on the activity of brain areas involved in working memory tasks and on performance in different cognitive domains [4,5]. It also improves executive functions, such as mental flexibility and set-shifting [6], to produce long-lasting beneficial effects when combined with conditioning tests [7].

Despite much evidence supporting the view of a general enhancement of cognition, the effects of food restriction on the central nervous system are still debated due to experimental differences that include statistical issues, diet composition, protocols and settings [4]. Potential applications of fasting or dietary restriction to the most common neurological diseases remain poorly investigated [8]. 

Animal studies, which provide mechanistic insights into the effects of food deprivation or restriction, mostly focused on the hippocampus for its involvement in cognitive functions and on the nucleus accumbens for its role in reward and motivation. Food restriction has mainly been tested as a tool to enhance hippocampal-dependent memory performance [9,10], leading to improvement in cognition and synaptic efficacy. In the hippocampal CA (Cornu Ammonis)1 area, short-term dietary restriction increases the number of glutamate receptors at the synapses responsible for an enhanced long-term potentiation (LTP) and α-Ammino-3-idrossi-5-Metil-4-idrossazol-Propionic Acid receptor (AMPAR) membrane incorporation [11,12]. Dietary restrictions are also associated with a reorganization of glutamatergic synapses, increasing surface expression and postsynaptic density abundance of GluA1 subunits of AMPARs, suggesting synaptic incorporation of these GluA2-lacking Ca^2+^-permeable AMPARs [13]. Other studies reported a compensatory upregulation of D1 Dopamine (DA) receptors [14,15] with enhanced phosphorylation of the glutamatergic AMPAR GluA1 at Ser845. This downstream effect increases peak current, facilitates the trafficking to the cell surface [16,17,18] and stabilizes the membrane Ca^2+^-permeable AMPARs [19]. In cells expressing these receptors, a combination of fast decay kinetics and large conductances that enhance synaptic transmission create the conditions for metaplasticity, in which the synapses become primed for a preferential direction of plasticity [20,21]. 

The dorsolateral part of the nucleus striatum hosts two forms of D1 DA-dependent synaptic plasticity that encode specific action-outcome associations in goal-directed behaviors. An intact function of striatal spiny projection neurons (SPNs) is needed for action selection and initiation through the integration of sensorimotor, cognitive, and motivational/emotional information [22]. Relevant to eating behaviors, the dorsal striatum is a site of action of DA control of motivation for food in both humans and animals [23,24]. 

Based on previous findings showing that food deprivation may impact glutamatergic synapses through enhancement of GluA2-lacking AMPARs-mediated activity in inhibitory neurons [25], we explored whether in SPNs of mice acutely deprived for 12 hours/day, for two consecutive days, we could retrieve signs of the synaptic insertion of GluA2-lacking AMPARs. Towards this aim, we analyzed the glutamatergic neurotransmission and the principal forms of DA-and glutamate-dependent synaptic plasticity: the corticostriatal long-term depression (LTD) and LTP. 

## 2. Results

### 2.1. Acute Food Restriction Protocol Induced Long-term Changes in Spontaneous Glutamatergic Synaptic Currents in the Corticostriatal Synapses

To explore the long-term effects of acute food restriction on the activity of striatal SPNs, we first analyzed the basal membrane properties through ex vivo patch-clamp and intracellular recordings from SPNs in corticostriatal slices obtained from 40-days-old Food Restricted (FR) and aged-matched C57BL/6J male mice (Naïve) (Figure 1A). The current-voltage relationship, obtained by applying hyperpolarizing and depolarizing current steps to SPNs, showed no significant differences between the two experimental groups. No differences were observed in the resting membrane potential (RPMs) and in the firing patterns (Figure 1B,C, RPM -85.89 ± 0.95 mV for Naïve and −87.54 ± 0.66 mV for FR; Student’s *t*-test *p* > 0.05, Naïve *n* = 18, FR *n* = 28). The firing rate was also unchanged as the mean number of spikes was similar between FR and Naïve mice (Figure 1D, 14.65 ± 0.51 *n* = 17 for Naïve and 16.00 ± 0.71 *n* = 15 for FR).

### 2.2. SPNs of FR Mice Showed Increased Inwardly Rectifying AMPARs Currents and Unbalanced AMPA:NMDA Ratio

In a separate set of experiments, in corticostriatal slices, we recorded combined AMPAR-N-methyl-D-aspartate receptor (NMDAR)–mediated excitatory postsynaptic currents (EPSCs) at +40 mV, and then to determine the rectification index (RI), we recorded AMPAR EPSCs at different holding potentials (−70 mV, 0mV, +40 mV) in the presence of D-(-)-2-Amino-5-phosphonopentanoic acid (D-APV, 50 μM), a selective NMDAR antagonist. In contrast with Naïve mice, we found that FR mice showed an increase in RI whose value differed significantly from the control mice (Figure 1E, current/voltage curve two-way ANOVA time x group interaction, Naïve *n* = 19 *vs.* FR *n* = 11, F_(2,56)_ = 5.36, *** *p* < 0.001; bar graph unpaired t-test, Naïve *n* = 13 *vs.* FR *n* = 15, *t* = 7.942, df = 28, *** *p* < 0.001). We then examined the AMPAR: NMDAR ratio at +40 mV and observed a significant difference between the two experimental groups (Figure 1F, unpaired t-test, Naïve *n* = 10 *vs.* FR *n* = 8, *t* = 2.911, df = 16, * *p* < 0.05). The decreased ratio in FR mice suggests a correlation between this parameter and the RI, indicating an increased contribution of the GluA2-lacking AMPAR to the EPSC. Notably, in SPNs of FR mice, AMPA-mediated currents showed a marked difference in kinetics, with a more rapid decay of EPSCs (Figure 1F, bottom right panel, green lines).

### 2.3. Enhanced GluA1-mediated Function in Striatal SPNs of FR Mice was associated with a Change in the Direction of Corticostriatal Synaptic Plasticity 

To explore if changes in AMPAR subunit composition could affect the corticostriatal glutamatergic transmission, we examined spontaneous EPSCs (sEPSCs) in SPNs of mice of the two experimental groups. As reported in Figure 2B, sEPSCs frequency was significantly increased in FR mice compared with Naïve mice (Figure 2A; unpaired t-test, Naïve *n* = 10, vs. FR *n* = 8, *t* = 3.088, df = 19, ** *p* < 0.01). Conversely, the amplitude of sEPSCs was comparable in SPNs between the two groups (Figure 2A). 

Subsequently, we tested the ability of SPNs to express the long-term depression (LTD). In control condition, in which the bathing solution contained a physiological concentration of magnesium ions, a high-frequency stimulation (HFS) protocol of the corticostriatal fibers induced a robust LTD of the excitatory postsynaptic potentials (EPSPs) in the SPNs of Naïve mice (Figure 2B, paired t-test pre- *vs.* 20 min post-HFS, Naïve *n* = 7, *t* = 12.10, df = 12, *** *p* < 0.001). In contrast, the induction of this form of synaptic plasticity was impaired in FR mice and, interestingly, we observed a long-term potentiation (LTP) (Figure 2B, paired t-test pre vs. 20 min post-HFS, FR *n* = 9, *t* = 7.299, df = 17, *** *p* < 0.001), resulting in a significant difference on the response to HFS between the two groups (Figure 2B, two-way ANOVA: time x group interaction F_(24,336)_ = 18.85, Bonferroni’s post hoc **^###^***p* < 0.001).

### 2.4. Selective GluA1 Antagonism was Associated with the Reappearance of Corticostriatal LTD in FR Mice

To test if this unexpected form of plasticity could depend on the increase in GluA1-mediated activity, we analyzed the LTD expression in the presence of a selective blocker of GluA2-lacking AMPARs, 1-naphthylacetyl spermine (NASPM, 30 μM). EPSPs were recorded for 10 min to obtain a stable baseline and then for 20 min after applying the HFS protocol. We found that in the presence of NASPM, SPNs recorded in Naïve and FR mice showed LTDs of similar amplitudes (Figure 2C, paired t-test pre *vs.* 20 min post-HFS, Naïve *n* = 8, *t* = 8.017, df = 15; FR *n* = 9, *t* =10.55, df = 17, *** *p* < 0.001 for both groups). Bath application of 30 μM NASPM in corticostriatal slices from FR mice efficiently reduced the EPSPs amplitude after HFS, contrasting the unphysiological potentiation observed in the untreated condition, demonstrating that blocking GluA2-lacking AMPARs prevents the shift in synaptic plasticity direction.

### 2.5. Enhanced GluA1-AMPARs Function was Associated with Changes in LTP Maintenance in Striatal SPNs of FR Mice

Since AMPARs play a variety of roles in shaping synaptic plasticity and are important for both LTD induction and LTP maintenance, we explored if the time course of LTP was also changed by using whole-cell patch-clamp recordings of SPNs in corticostriatal slices. To study this form of plasticity, Mg^2+^ ions were removed from the medium to promote the activation of glutamate NMDARs.

Under these experimental conditions, an initial post-tetanic potentiation was normally induced by the application of an HFS protocol in both Naïve and FR mice. Although comparisons between EPSP amplitudes before and 20 minutes after HFS indicates that a slight potentiation could still be observed in FR mice (paired t-test pre *vs*. 20 min post-HFS, Naïve *n* = 5, *t* = 12.58, df = 9, *p* < 0.0001, FR *n* = 8, *t* = 4.075, df = 15, *p* < 0.001), the LTP maintenance was different in the two groups. In FR mice, the amplitude of EPSPs decreased over time, bringing to a significant difference in the strength of LTP between the two groups (Figure 3, two-way ANOVA: time × group interaction F_(24,264)_ = 4.23, 11–20 min, Bonferroni’s post hoc # *p* < 0.05).

## 3. Discussion

In this paper, we present evidence that acute but moderate (12 hours/day, for two consecutive days) food restriction can be related to persistent and subunit-specific enhancement in AMPARs function in SPNs of the dorsolateral striatum that emerges under specific stimulation of the corticostriatal pathway. 

In our experiments, striatal SPNs of FR mice displayed intrinsic membrane properties and amplitude of spontaneous glutamatergic-mediated activity comparable with ad libitum fed controls. However, we observed significant changes in other aspects of glutamatergic transmission with substantial modifications in the glutamate-dependent synaptic plasticity linked to an enhanced function of GluA2-lacking AMPARs. 

According to findings showing that food deprivation is associated with an enhanced abundance of Ca^2+^-permeable AMPARs at glutamatergic synapses [13], here we describe a significant increase of the AMPAR rectification index. In our paradigm, this measure provided an indication of the changed proportion of the GluA1 over the GluA2 AMPAR subunits. These latter are the most expressed subunits of AMPARs in the striatum of adult rodents [26,27,28], characterized by a unique editing at the mRNA level where a glutamine codon is switched to arginine that confers channel resistance to Ca^2+^. As a result, the neurons with increased insertion of homomeric GluA1 AMPARs show inward rectification that becomes linear when the GluA1 subunits are coexpressed with the GluA2 subunits [29,30]. Therefore, AMPAR complexes that lack GluA2 are permeable to sodium and Ca^2+^ ions, resulting in a higher single channel conductance and fast decay kinetics. 

Since striatal AMPARs and NMDARs act in concert with dopaminergic neurotransmission to shape the direction of corticostriatal synaptic plasticity, we explored a possible imbalance between AMPA- and NMDA-mediated glutamatergic transmission to find a link between the different AMPAR subunit composition and the changes in the AMPA:NMDA ratio. Our data show a reduction of such a ratio in SPNs of Naïve and FR mice and, as confirmation of the increased GluA1-mediated activity, AMPAR-mediated currents were markedly different in their decay slope, exhibiting faster channel deactivation kinetics [31] and more rapidly decaying EPSPs [32]. 

We then tested the hypothesis that in deprived mice, the corticostriatal LTD, which depends on the activation of AMPARs [33], was also changed.

In SPNs recorded from FR mice, a high-frequency stimulation protocol of the corticostriatal fibers that, in physiological condition, elicited LTD, induced a shift toward LTP. This form of synaptic plasticity, whose induction is typically dependent on NMDAR activation, was not observed at physiological concentrations of magnesium ion, which acts as a natural blocker of the receptor pore. Such a change in synaptic plasticity direction might depend on increased Ca^2+^ entrance due to the enhancement of GluA2-lacking activity after a strong afferent stimulation. In such conditions, a massive corticostriatal stimulation would produce a greater depolarization of the SPNs membrane, relieving the NMDA receptors from the magnesium block and facilitating the induction of LTP. 

This interpretation was substantiated by the present electrophysiological findings showing that NASPM, a selective blocker of GluA1-bearing AMPARs, restored a physiological LTD in FR mice at a dose that did not induce any effects in Naïve mice.

A possible explanation for the effect of such a moderate restriction protocol could be sought in a coincident impact of food deprivation on AMPARs changes during the development. In the newborn striatum, GluR1 immunoreactivity was observed in the presynaptic neurites, forming synapses with a more pronounced presence at the postsynaptic level in morphologically mature synapses as shown in seminal immunoelectron microscopy studies [34]. Moreover, the expression of GluA2-lacking AMPA receptors at excitatory synapses have been detected in many brain regions in the early postnatal development [35], and the switch from this subunit to the GluA2-containing AMPARs subunit occurs within the first two to three postnatal weeks [36,37,38]. However, we tested the animals far beyond these time points, when subunit composition had reached a steady-state with a net prevalence of GluA2-containing AMPARs. Thus, we excluded possible confusing developmental factor’s effect on corticostriatal synaptic activity.

Given that AMPARs govern a variety of functions, including a fine regulation of both Ca^2+^ influx and LTP maintenance, and the link with Ca^2+^-mediated kinase II [17,18,39], we explored the possibility that a change in their function could also affect the maintenance of LTP. While LTP was typically induced in SPNs from both groups, the amplitude of LTP in FR mice degraded over time. 

Although SPNs show a peculiar pharmacological modulation of LTP, due to co-activation of DA and glutamate receptors and concurrent modulation by interneuronal activity, a possible explanation would be that the phase transition between induction and maintenance of LTP requires a change in membrane insertion of AMPARs subunits, as observed in the hippocampal CA1 area. In fact, in pyramidal neurons of CA1, during LTP induction, an initial incorporation of GluA2-lacking Ca^2+^-permeable AMPARs is followed by a replacement with GluA2-containing Ca^2+^-impermeable receptors [40]. However, given that this aspect has not been investigated in striatal SPNs, additional analyses are required to clarify these dynamics and the relevant contribution of AMPAR- and NMDAR-mediated components in the reduction of LTP.

These results support the view that a sudden, although moderate, food deprivation in experimental animals that were fed ad libitum since birth could be a salient event that may be encoded into a series of synaptic adaptations associated with long-term effects with adaptive changes in the AMPAR-mediated functions. These changes observed in isolated currents also emerged upon electrical stimulation of afferents without affecting the NMDAR-dependent phases of corticostriatal plasticity. This is in agreement with other studies reporting that increased Ca^2+^ influx via AMPARs lacking the GluA2 subunit does not have an impact on the NMDA component of LTP [41] and might be managed in homeostatic conditions.

Further investigations should clarify if the changes in AMPARs subunit composition are only limited to the postsynaptic level. In fact, presynaptic GluA1-AMPARs have been identified in corticostriatal and thalamostriatal axon terminals [42,43]. A possible increased insertion of these receptors [13] may explain the increased sEPSC frequency observed in our experimental setting. 

A concept of AMPA-dependent changes in presynaptic activity has already been put forward in past studies using in vivo microdialysis and showing enhanced release of glutamate in the striatum upon perfusion of AMPA that was blocked by AMPA antagonists [43], an effect associated with presynaptic adenylyl cyclase-dependent processes [44]. These data are in agreement with more recent findings demonstrating that AMPARs localize at presynaptic sites on glutamatergic afferents [42], and that AMPA autoreceptors would assure a positive feedback control of glutamate release that modulates synaptic scaling with nonlinear characteristics [42]. This system would work in balance with presynaptic metabotropic glutamate receptors, which, on the contrary, suppress or inhibit the release from axon terminals [45]. 

Relevant to our observations, these findings suggest that a dynamic regulation of synaptic scaling might also occur in FR mice, where an LTP is observed instead of an LTD. Given the increase of sEPSC frequency, this shaping in the direction of plasticity would be not only associated with an adaptive enhancement of postsynaptic GluA2-lacking-mediated function (resulting in an increase of rectification index) but also extend to changes in the AMPAR activity at presynaptic levels. A point that still awaits to be clarified is the identification of the duration of possible adaptive changes in AMPAR subunits composition in cortico- and thalamostriatal afferents, and if a switch toward GluA2-lacking subunit-dependent function can also be detected at presynaptic sites.

In conclusion, with the present electrophysiological results, this study provides new insights into the importance of Ca^2+^-permeable GluA1 AMPARs and their involvement in the two primary forms of striatal plasticity, LTD and LTP. Dissecting the role of GluA2-lacking AMPARs, although less expressed in the adult striatum, can be critical for a full understanding of the mechanisms of compensative synaptic regulations that may occur in physiological conditions when the limits of homeostasis are challenged. 

## 4. Materials and Methods

### 4.1. Animals

Male C57BL/6JO1aHsd mice (*n* = 23; approximately 40 days old at study onset) (Harlan, Italy) were used. All animals were housed four per cage, under a controlled 12-h light/12-h dark cycle and temperature (22–23 °C), with food and water ad libitum. All efforts were made to minimize the number of animals used and their suffering, in accordance with the European Directive (2010/63/EU). All procedures were approved by the institutional review board and ethics committee (IRCCS Fondazione Santa Lucia) and by the Italian Ministry of Health (Project identification code: 534/2019-PR). The animals were randomly allotted into two groups, Food Restricted (FR) and aged-matched C57BL/6J male mice (Naïve). The FR mice were subjected to a moderate food deprivation protocol that limited their access to food for 12 hours, during the dark phase, for two consecutive days [46]. Such a regimen resulted in no significant body weight loss. Thirty days later, both Naïve and FR mice were sacrificed for the electrophysiological recordings (Figure 1A). 

### 4.2. Slices Preparation

FR male mice and Naïve aged-matched C57BL/6J male mice were used in electrophysiological experiments. All mice were sacrificed by cervical dislocation and the brain was rapidly removed from the skull. Corticostriatal slices were cut from mice brains (thickness, 240–280 µm) using a vibratome in Krebs’ solution (in mmol/L: 126 NaCl, 2.5 KCl, 1.2 MgCl_2_, 1.2 NaH_2_PO_4_, 2.4 CaCl_2_, 10 glucose, and 25 NaHCO_3_) bubbled with a 95% O_2_–5% CO_2_ gas mixture. After at least 1 hr recovery, individual slices including the cortex and the striatum were transferred to a recording chamber and continuously superfused with oxygenated Krebs’ medium, at 2.5–3 mL/min and maintained at 32–33 °C. 

### 4.3. Whole-cell Patch-clamp Recordings

Current-clamp recordings from spiny projection neurons (SPNs) were performed using the whole-cell patch-clamp technique. Neurons of the dorsal striatum were visualized using infrared differential interference contrast microscopy (Eclipse FN1, Nikon, Tokyo, Japan) [47,48]. Recordings were made with a Multiclamp 700B amplifier (Molecular Devices, San José, CA, USA) and stored on PC using pClamp 9 (Molecular Devices, San José, CA, USA). Borosilicate glass pipettes (6–9 MΩ) were filled with the following internal solutions (in mM): 120 K-gluconate, 0.1 CaCl_2_, 2 MgCl_2_, 0.1 EGTA, 10 N-(2-hydroxyethyl)-piperazine-N-s-ethanesulfonic acid, 0.3 Na-guanosine triphosphate, and 2 Mg-adenosine triphosphate (Mg-ATP), adjusted to pH 7.3 with KOH. For recordings in voltage-clamp mode the internal solution of glass pipettes contained (in mM): 120 CsMeSO_3_, 10 CsCl, 8 NaCl, 2 MgCl_2_, 10 HEPES, 0.2 EGTA, 10 TEA, 5 QX314, 0.3 NaGTP and 2 Mg-ATP. Striatal neurons were clamped at the holding potential of -80 mV and identified by the absence of spontaneous action potential discharge. 

Whole-cell access resistance was 15–30 MΩ. Picrotoxin (50 µM) was added to block GABA_A_-currents. D-APV (50 µM), an N-methyl-D-aspartate receptor (NMDAR) antagonist, was used to pharmacologically isolate the Ammino-3-idrossi-5-Metil-4-idrossazol-Propionic Acid receptor (AMPAR)-excitatory postsynaptic currents (EPSCs). To obtain the NMDA current, an evoked current at +40 mV was subtracted before and after the application of this antagonist. To calculate the AMPAR:NMDAR ratio, the AMPAR EPSC amplitude was divided by the NMDAR EPSC amplitude, both measured at +40 mV. The rectification index (RI) of AMPARs was obtained by dividing the chord conductance calculated at negative potential (−70 mV) and positive potential (+40 mV). Injected currents and input resistances were checked throughout the experiments. Cells with variations in these parameters >20% were rejected.

### 4.4. Intracellular Recordings with Sharp Electrodes

Current-clamp recordings, with an intracellular technique, were performed blindly using sharp electrodes filled with 2 M KCl (30−60 MΩ). Signal acquisition was performed with an Axoclamp 2B amplifier (Molecular Devices, San José, CA, USA), displayed on a separate oscilloscope, stored. Online and offline analyses were performed using a digital system (pClamp 9, Molecular Devices, San José, CA, USA). 

To evoke EPSCs and glutamatergic excitatory postsynaptic potentials (EPSPs), the stimulating bipolar electrode and the recording electrodes were located in the white matter and within the dorsolateral striatum, respectively.

In both patch-clamp and intracellular recordings, EPSPs were evoked by electrical stimulation every 10 s and EPSP peak amplitudes were used as a measure of SPNs activity, according to previous studies from our and other research groups, which consider this parameter an optimal index to evaluate the extent of evoked striatal responses when recording in current-clamp mode [33,49,50,51,52]. To induce long-term depression (LTD) and long-term potentiation (LTP), we used a high-frequency stimulation (HFS) protocol, consisting of three trains of 100 Hz, 3 s of duration, and 20 s of interval. For the LTP protocol, magnesium (Mg^2+^) ions were omitted from the Kreb’s solution to remove the Mg^2+^-dependent block of NMDA receptors [49]. During tetanic stimulation, the intensity of stimulation was increased to suprathreshold levels. EPSP modifications induced by HFS protocol were expressed as a percentage of control, the latter representing the mean of responses recorded during a stable period (15–20 min) before the tetanic stimulation. Current-voltage relationships were obtained by applying steps of current of 200 pA in both hyperpolarizing and depolarizing direction (from −400 to +200 pA). Firing frequency was calculated as the mean number of spikes in response to a step of 600 pA and shown as averaged values in scatter dot plots. 

### 4.5. Chemicals

D-(-)-2-Amino-5-phosphonopentanoic acid (D-APV) was from Tocris Bioscience (Bristol, U.K.); 1-naphthylacetyl spermine trihydrochloride (NASPM) and Picrotoxin were from Sigma-Aldrich (Milan, Italy). Drugs dissolved in the final concentration were bath applied by switching the control perfusion to drug-containing solution. During the cell’s recording, an aliquot of the stock APV solution was diluted in Krebs’ solution to 50 μM and kept in a syringe for the entire duration of the experiment. For the NASPM and the Picrotoxin an aliquot of the stock solution was diluted in Krebs’ solution to 30 μM and 50 μM, respectively. 

### 4.6. Statistical Analyses

Analyses were performed using Prism 6.0 (GraphPad software). Electrophysiological results are presented as mean ± SEM. Paired Student’s *t-*test was used for analysis of the mean pre vs post-HFS in the same cell population. Analysis of variance (ANOVA) test with a post hoc Bonferroni test were performed among different neuronal populations. Sample size was calculated with G*Power software (5% type I error; 80% power). 

## Figures and Tables

**Figure 1 ijms-22-01916-f001:**
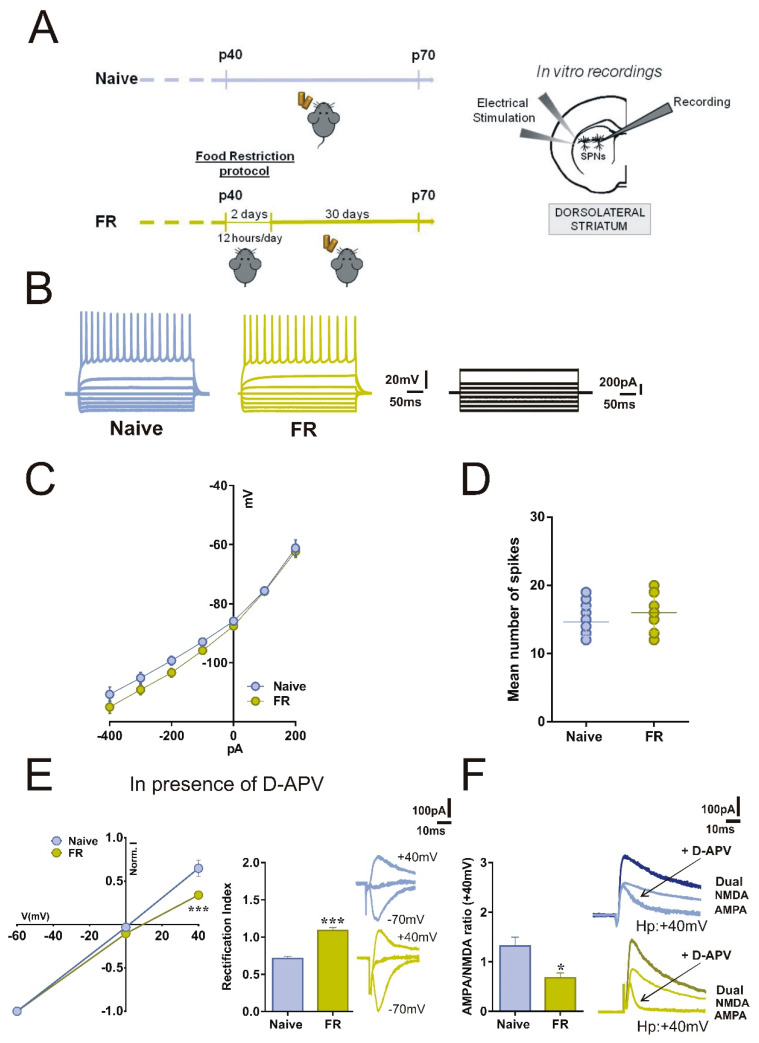
Acute food restriction is associated with changes in glutamatergic transmission in spiny projection neurons (SPNs). (**A**) Experimental plan of Naïve and food-restricted condition. (**B**) Representative firing traces and (**C**) current-voltage (I/V) graphs of Naïve (*n* = 17) and FR mice (*n* = 15), obtained after applying hyperpolarizing and depolarizing steps of current to SPNs recorded in dorsolateral striatum. (**D**) The aligned dot plot shows the mean number of spikes triggered by a step that generates a maximum response. (**E**) Current/voltage curve and bar graph show the rectification pattern and the rectification index in SPNs of Naïve and FR mice. The α-Ammino-3-idrossi-5-Metil-4-idrossazol-Propionic Acid receptor (AMPAR)- excitatory postsynaptic currents (EPSCs) were pharmacologically isolated by application of the N-Methyl-d-aspartate (NMDAR) antagonist D-(-)-2-Amino-5-phosphonopentanoic acid (D-APV, 50 µM) (current/voltage curve, two-way ANOVA time x group interaction, Naïve *n* = 19 *vs*. FR *n* = 11, F_(2,56)_ = 5.36, *** *p* < 0.001; bar graph, unpaired t-test, Naïve n = 13 *vs*. FR *n* = 15, *t* = 7.942, df = 28, *** *p* < 0.001). Example traces of evoked AMPAR-EPSCs recorded at -70, 0, and+40 mV. Scale bar: 100 ms, 100 pA. (**F**) Group mean AMPA:NMDA ratio calculated in Naïve and FR SPNs in the presence of D-APV (unpaired t-test, Naïve *n* = 10, *vs*. FR *n* = 8, *t* = 2.911, df = 16, * *p* < 0.05); example traces of evoked AMPA- and NMDA-EPSCs at +40 mV (Dual: AMPA + NMDA EPSCs; NMDA EPSCs: obtained by subtraction of the EPSCs measured before and after the application of 50 µM D-APV; AMPA EPSCs: isolated by application of 50 µM D-APV). Scale bar: 100 ms, 100 pA.

**Figure 2 ijms-22-01916-f002:**
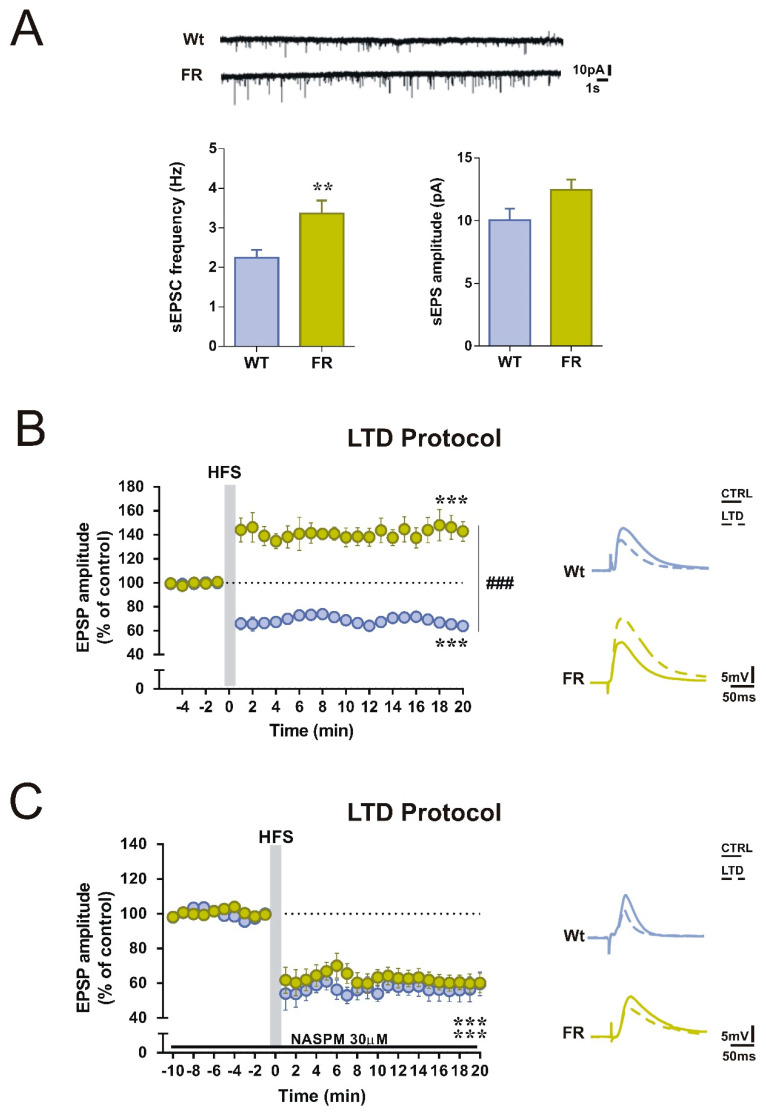
Enhanced activity of GluA2-lacking AMPARs in food-restricted (FR) mice is associated with changes in the direction of corticostriatal synaptic plasticity in SPNs. (**A**) Frequency and amplitude of sEPSCs glutamatergic transmission in Naïve and FR mice. In the upper part, comparison traces of spontaneous activity recorded from groups are shown. The frequency of sEPSC is increased in FR mice compared to Naïve mice (unpaired t-test, Naïve *n* = 10, *vs*. FR *n* = 8, *t* = 3.088, df = 19, ** *p* < 0.01). (**B**) Left panel: time course of excitatory postsynaptic potential (EPSP) amplitude of SPNs from Naïve and FR mice after induction of long-term depression (LTD) protocol (high-frequency stimulation, HFS) (paired t-test pre *vs*. 20 min post-HFS, Naïve *n* = 7, *t* = 12.10, df = 12, *** *p* < 0.001, FR *n* = 9, *t* = 7.299, df = 17, *** *p* < 0.001). Grouped analysis shows significant group effect (two-way ANOVA: time x group interaction F_(24,336)_ = 18.85, Bonferroni’s post hoc ^###^
*p* < 0.001). The scale factor is 50 ms/5 mV for all traces. Right panel, representative traces of single SPNs recorded from Naïve and FR mice before (solid lines) and after HFS (dotted lines). (**C**) Left panel: time course of SPNs EPSP amplitude, recorded from Naïve and FR mice in the presence of 30 μM 1-naphthylacetyl spermine (NASPM) bath application for the whole duration of the experiment (paired t-test pre *vs*. 20 min post-HFS, Naïve *n* = 8, *t* = 8.017, df = 15; FR *n* = 9, *t* = 10.55, df = 17, *** *p* < 0.001 for both groups). The scale factor is 50 ms/5 mV for all traces. Right panel, representative traces of single SPNs recorded from Naïve and FR mice before (solid lines) and after HFS (dotted lines).

**Figure 3 ijms-22-01916-f003:**
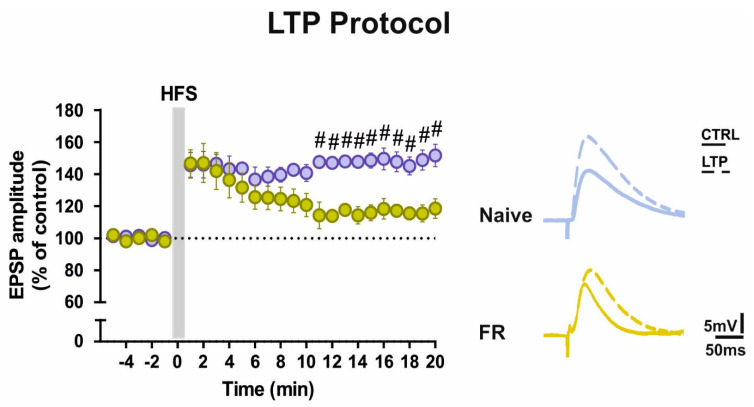
AMPAR subunit composition is critical for the maintenance of long-term potentiation (LTP). Left panel: time course of EPSP amplitude of SPNs from Naïve and FR mice after induction of LTP protocol. Grouped analysis shows significant group effect (two-way ANOVA: time x group interaction F_(24,264)_ = 4.23, 11–20 min, Bonferroni’s post hoc ^#^
*p* < 0.05). The scale factor is 50 ms/5 mV for all traces. Right panel: representative traces of single SPNs recorded from Naïve and FR mice before (solid lines) and after HFS (dotted lines).

## Data Availability

The data presented in this study are available on request from the corresponding author.

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
