# Peer review of "Long-Term Shaping of Corticostriatal Synaptic Activity by Acute Fasting"

_ijms, 2021, doi:10.3390/ijms22041916_

Round 1

Reviewer 1 Report

Long-term shaping of corticostriatal synaptic activity by acute fasting

This is a well described study in addressing food restriction and the effect on glutamateric synapses. The focus of wither AMPARs are altered is an important topic to address. The study clearly demonstrated changes which would suggest an enhancement in AMPARs in the dorsolateral striatum with food restriction. The text is well written and easy to follow. The figures are clear and understandable.

There is only one major point that I am not clear on. This relates with Figure 2A.

The authors state,  " The frequency of sEPSC is increased in FR mice compared to Naïve mice (unpaired t-test, Naïve n = 10, vs. FR n = 8, t = 3.088, df = 19, **p < 0.01)"

So, it is interesting on the increase in the amplitude which as stated can relate to the Mg plug being removed and more AMPA receptors being present. But what is not clear to me is why is the frequency of spontaneous EPSCs increased with FR. This would suggest a presynaptic alteration would it not ?

There does not appear to be much discussion on the frequency changes and potential of presynaptic contributions to this point. Can this be addressed why or why not there could be a presynaptic alteration with FR?

Author Response

We thank the Reviewer for asking about this important point that we have just mentioned in the original manuscript at the end of the discussion section, where we stated that further investigations are needed to clarify possible presynaptic changes caused by the switch in AMPAR subunits composition. We are now more motivated to further elaborate on this point and according to the Reviewer’s advice we improved the discussion of the data on the sEPSC frequency increase at page 9, lines 286-307:

“A concept of AMPA-dependent changes in presynaptic activity has already been put forward in past studies using in vivo microdialysis and showing enhanced release of glutamate in the striatum upon perfusion of AMPA that was blocked by AMPA antagonists (Patel et al., 2001), an effect associated with presynaptic adenylyl cyclase-dependent processes (Dohovics et al., 2003). These data are in agreement with more recent findings demonstrating that AMPARs localize at presynaptic sites on corticostriatal and thalamostriatal glutamatergic afferents (Fujiyama et al., 2004). The authors state that the presence of AMPA autoreceptors would assure a positive feed-back control of glutamate release that modulates synaptic scaling with non-linear characteristics (Fujiyama et al., 2004), a system that would work in balance with pre-synaptic metabotropic glutamate receptors, which, on the contrary, would suppress or inhibit the release from axon terminals (Rouse et al., 2000).

Relevant to our observations these findings suggest that a dynamic regulation of synaptic scaling might also occur in our paradigm, where a GluA1-dependent LTP is observed instead of LTD. This shaping of plasticity direction would be then not only associated with an adaptive enhancement of postsynaptic GluA1-mediated function - resulting in an increase of rectification index, although only a trend on increased amplitude can be observed – but also extend to presynaptic levels accounting for the in-crease in frequency. Further investigations are needed to clearly identify the nature of food-deprivation-induced changes in AMPAR subunits composition in cortico- and thalamo-striatal afferents and if the switch toward GluA1 subunit-dependent function can be detected also at presynaptic sites.”

Reviewer 2 Report

Major comments:

  1.  The authors cited previous studies [9, 10, 11] to demonstrate that GluA1 subunits of AMPA play an important role in food restriction mediated reorganization of glutamatergic synapses. Male adults rats (minimum body weight 350g, at least 10 weeks old https://www.taconic.com/pdfs/sprague-dawley-rat.pdf) were used in all these studies. Developmental factor could influence the synaptic AMPA receptor distribution and AMPA receptor subunit composition. In this study, the authors use adolescent mice (5 weeks old) and waited 30 days (10 weeks old) for further investigation. How could the authors exclude the possibility of developmental factor's effect on corticostriatal synaptic activity?
  2. " The decreased ratio in FR mice suggests a correlation between this parameter and the RI, indicating an increased contribution of the GluA1 over the GluA2 AMPAR subunits". Without direct evidence, the authors suggested that GluA1 was elevated. This is not supportive and convincible. Immunoblot analysis should be performed to confirm this conclusion as pervious study did [9]. 

Author Response

a. We thank the Reviewer for raising this interesting point that should definitely be mentioned in the discussion. We are aware that, as suggested, developmental factors may influence the AMPA-dependent corticostriatal synaptic activity and we cannot exclude a priori this possibility. In fact, both synaptic localization and AMPARs subunit composition are subjected to developmental changes. In the newborn striatum, GluR1 immunoreactivity was observed in the presynaptic neurites forming synapses with more pronounced presence at postsynaptic level in morphologically mature synapses as shown in seminal immunoelectron microscopy studies (Martin et al., 1998). Moreover, the expression of GluA2-lacking AMPA receptors at excitatory synapses have been detected in many brain regions in the early postnatal development (Kumar et al., 2002), and the switch from this subunit to the GluA2-containing AMPARs subunit occurs within the first 2–3 postnatal weeks (Eybalin et al., 2004, Ho et al., 2007; Bellone et al., 2011). However, we tested the animals far beyond these time points, when subunit composition has reached a steady state with a net prevalence of GluA2-containing AMPARs.

Thus, these previous findings and the fact that we did not observe similar changes in naïve mice (fed ad libitum) would make the possibility of developmental factor's effect on corticostriatal synaptic activity unlikely. Nevertheless, we recognize that the relationships between developmental differences in striatal neuron ion channel properties and the expression of specific AMPA and NMDA GluR subtypes remain to be evaluated. A discussion on this point has been now included in the revised version of the manuscript at page 8/15, lines 245-258:

“A possible explanation for the effect of a such moderate restriction protocol could be sought in a coincident impact of food deprivation on AMPARs changes during the development. In the healthy brain, both synaptic localization and AMPARs subunit composition are, in fact, subjected to developmental changes. In the newborn striatum, GluR1 immunoreactivity was observed in the presynaptic neurites forming synapses with more pronounced presence at postsynaptic level in morphologically mature syn-apses as shown in seminal immunoelectron microscopy studies (Martin et al., 1998). Moreover, the expression of GluA2-lacking AMPA receptors at excitatory synapses have been detected in many brain regions in the early postnatal development (Kumar et al., 2002), and the switch from this subunit to the GluA2-containing AMPARs subu-nit occurs within the first 2–3 postnatal weeks (Eybalin et al., 2004, Ho et al., 2007; Bellone et al., 2011). However, we tested the animals far beyond these time points, when subunit composition has reached a steady state with a net prevalence of GluA2-containing AMPARs. Thus, we excluded possible confusing developmental factor's effect on corticostriatal synaptic activity”.

b. We thank the Reviewer for asking this critical question. The molecular quantification of GluA1 subunit is an interesting point to explore. Nevertheless, exposure to this restriction protocol did not alter intrinsic membrane properties nor induced a full impairment but rather a change in the direction of plasticity, and mice phenotype is also unchanged (data not shown). We therefore assume that the observed electrophysiological features may occur in the absence of detectable molecular changes in GluA1 expression, as molecular and cellular AMPARs trafficking events that control synaptic responsiveness and plasticity, are governed by a complex series of intracellular pathways, whose changes are often hard to detect even in pathological conditions at stages in which, however, early synaptic changes are observed. However, we think that an analysis of GluA1 expression would be definitely be included in future experiments on the molecular changes associated to different protocols of food restriction.

Round 2

Reviewer 2 Report

The authors adequately answered my concerns. 

Author Response

We thank the Reviewer for her/his answer